# Fish Oil Replacement by a Blend of Vegetable Oils in Diets for Juvenile Tench (*Tinca tinca* Linnaeus, 1758): Effects on Growth Performance and Whole-Body Composition

**DOI:** 10.3390/ani12091113

**Published:** 2022-04-26

**Authors:** María Sáez-Royuela, Teresa García, José M. Carral, Jesús D. Celada

**Affiliations:** Departamento de Producción Animal, Facultad de Veterinaria, Universidad de León, Campus de Vegazana s/n, 24071 León, Spain; etcmtg03@estudiantes.unileon.es (T.G.); jmcarl@unileon.es (J.M.C.); jdcelv@unileon.es (J.D.C.)

**Keywords:** sustainable diets, fish oil replacement, vegetable oils, juvenile tench rearing, growth performance, whole-body fatty acid composition

## Abstract

**Simple Summary:**

Fish oil (FO) can be totally replaced by a blend of vegetable oils (30% linseed oil, 20% corn oil, and 50% olive oil) in diets for juvenile tench (*Tinca tinca* Linnaeus, 1758) without negative effects on survival rate and growth performance. Diets affected the composition of juveniles so that lipid content was significantly lower in animals fed diets without FO. Moreover, the content of linolenic acid (18:3n-3) increased as FO substitution did, being significantly higher with respect to control diet (without vegetable oils) from 40% or higher FO replacement diets. No differences in saturated, monounsaturated, and polyunsaturated fatty acids were found in the fish whole-body. Nutritional indices can be considered within optimal values for healthy foods.

**Abstract:**

Among freshwater species, tench (*Tinca tinca* Linnaeus, 1758) is considered as a promising species for the diversification of aquaculture, but the intensification of techniques is necessary to promote and consolidate its culture. Adequate feeding in early growth phases is essential to face further grow-out. Fish oil (FO) is the main source of lipids in fish diets, but its production is unsustainable, and thus, alternative oils should be considered. A 90-day experiment was performed to evaluate the effects of partial and total replacement of cod liver oil (FO) by a blend of vegetable oils (VO) in juvenile tench. Six isonitrogenous and isolipidic diets with different levels of a VO blend containing 30% linseed oil, 20% corn oil, and 50% olive oil were tested: 0% (control), 20%VO, 40%VO, 60%VO, 80%VO, and 100%VO. With all diets, survival was 100%, and there were not differences in growth performance (total length (TL); weight (W); specific growth rate (SGR); feed conversion ratio (FCR); and biomass gain (BG)). Compared to the control group, whole-body lipid content decreased significantly in the 100%VO group. No differences in total saturated (SFA), monounsaturated (MUFA), and polyunsaturated (PUFA) fatty acids were found in the whole-body. The content of linolenic acid (ALA) in the whole-body increased as FO substitution did, being significantly higher with respect to control diet from 40% FO replacement diets. Nutritional indices, such as ΣPUFA/ΣSFA and Σn-6/Σn-3 ratios, tended to increase with increasing VO content, whereas the EPA + DHA showed an opposite trend. A total replacement of FO by the blend of VO did not affect the growth performance and fatty acid profile of juvenile tench. Further research on the effects of VO diet on nutritional quality in tench reared to commercial size should be performed.

## 1. Introduction

Aquaculture plays a key role to achieve many of the United Nations’ sustainable development goals in 2030 [1]. However, the production of aquatic species often requires inputs from wild fish stocks in the form of feed ingredients [2]. The environmental footprint derived from the use of fishmeal (FM) and fish oil (FO) in feedstuffs is still a main concern to reach sustainable growth [3]. According to Naylor et al. [4], global landing of forage or small pelagic fish used to produce FM and FO have trended downward, whereas production of aquafeeds tripled between 2000 and 2017 [5]. Although alternative ingredients have been considered and, in some cases, included for full or partial substitution in feed formulation, at present, the aquaculture sector is still the biggest consumer, with a global use of 78% and 68% of FM and FO, respectively, in 2019 [6]. The high dependency on both ingredients is derived from the increase of aquaculture production, which will require an additional production of 37.4 million tonnes of aquafeeds by 2025 [7], but also by supplying essential nutrients to support larval and fry performance and survival [4]. In addition to its high digestibility, FO is a major natural source of some essential n-3 long chain polyunsaturated fatty acids (n-3 LC-PUFA), such as eicosapentaenoic acid (EPA) and docosahexaenoic acid (DHA) for finfish. Considering the low efficiency in the FO production process (one ton of pelagic fishes is needed to obtain an average of 50 kg of oil [8], and a huge deficit of FO supply is expected [9]). Facing this unsustainable scenario, it is necessary to find alternative lipid sources in the foreseeable growing aquafeed demand.

The largest and most widely used oil alternatives come from terrestrial plants, as production of vegetable oils (VO), in general terms, is more stable [10], and prices are lower than those for FO [11]. Lipids from vegetable sources are usually rich in C18 fatty acids, mainly linoleic (LA, 18:2n-6), α-linolenic (ALA, 18:3n-3), and oleic (OA, 18:1n-9), but they lack or have a very limited content of the n-3 LC-PUFA, such as EPA (20:5n-3) and DHA (22:6n-3) [12]. A large variety of plant oils, alone or in blends, have been considered as FO substitutes for many aquaculture species [10,13], highlighting soybean, palm, linseed, sunflower, rapeseed, and olive oils.

According to Torcher [14], partial or full substitution of FO is more feasible for freshwater than marine fish species, which apparently lack the ability to desaturate and elongate C-18 PUFA, and, therefore, are more likely to have n-3 LC-PUFA deficiency. Thus, most findings reported that FO replacement by VO does not have a negative effect on the growth performance of freshwater species, including rainbow trout (*Onchorhynchus mykiss*) [15,16,17,18,19], Arctic charr (*Salvelinus alpinus*) and brook charr (*S. fontinalis*) [20], Manchurian trout (*Brachymystax lenok*) [21], pikeperch (*Sander lucioperca*) [22,23], largemouth bass (*Micropterus salmoides*) [24], Beluga sturgeon (*Huso huso*) [25], silver barb (*Puntius gonionotus*) [26], mandarin fish (*Siniperca scherzeri*) [27], Nile tilapia (*Oreochromis niloticus*) [28,29,30], hybrid tilapia (*Oreochromis* spp.) [31], black carp (*Mylopharyngodon piceus*) [32], the minnow (*Onchystoma macrolepis*) [33], and common carp (*Cyprinus carpio*) [34,35,36]. Nevertheless, final fish composition, as whole-body or fish fillet, usually reflects feed composition, and several studies have shown that substitution of FO by VO can modify the healthiness properties and quality of fish by reducing their n-3 LC-PUFA content [10,37]. For this reason, it is necessary to evaluate the changes in the fatty acid profile that can compromise the healthiness properties of the fish and the benefits associated with its consumption.

Tench (*Tinca tinca* Linnaeus, 1758) is highly appreciated in many European countries, mainly for its gastronomic quality, but also as an attractive sport fishing species [38,39]. Although it is considered one of the most promising species for diversification of inland culture [40], its production is based on extensive culture systems where growth is often limited, leading to, usually, low and unpredictable yields [39,41]. For this reason, the development of intensification techniques is necessary to promote and consolidate tench production. To set up rearing procedures under controlled conditions, is essential to establish adequate feeding in early growth phases. As the availability of a specific diet is essential to face nutritional studies, García et al. and González-Rodríguez et al. [38,42] proposed a practical diet for juvenile tench, which allowed for good juvenile survival and growth performance. Later, Sáez-Royuela et al. [43] determined that 8.5% dietary lipid content (2% of cod liver oil) should be in the formulation. As there is no available information on alternative lipid sources in practical diet for this species, the aim of the present work was to evaluate the effects of partial or total replacement of FO by a mixture of vegetable oils on survival rate, growth performance, and whole-body composition of juvenile tench. Under the hypothesis that a blend of vegetable oils could provide a composition of essential fatty acids to cod liver oil, three oils were chosen: linseed oil as high source of ALA; corn oil, which provides high quantities of LA; and olive oil, rich in MUFA.

## 2. Materials and Methods

### 2.1. Fish, Facilities, and Experimental Procedure

The experiment was carried out in the indoor aquaculture facilities of the Department of Animal Production located in the Faculty of Veterinary of León (Spain). Tench larvae were obtained by hatching using artificial reproduction techniques [44], and reared in outdoor tanks. After four months, 540 juvenile tench from a homogenous pool were randomly distributed as groups of 30 fish in 18 fiberglass tanks (0.5 × 0.25 × 0.25 m) containing 25 L of water to obtain replicates corresponding to the different feeding treatments. Prior to distribution, 100 juveniles were anesthetized with tricaine methanesulfonate (MS-222; Ortoquímica S.L., Barcelona, Spain) to measure initial total length (TL) and weight (W). TL was measured with a digital caliper Sylvac (to the nearest 0.01 mm). After removing excess water with tissue paper, W was determined by a precision balance Cobos M-150-SX (to the nearest 0.001 g). Values of 23.87 ± 0.25 mm total length and 0.15 ± 0.05 g weight (mean ± mean standard error) were obtained. The total biomass of each tank was weighted. Following a monofactorial design, diet was the experimental factor with 3 replicates per level of treatment (three tanks with 30 juvenile tench per each experimental diet).

At the beginning of the experiment, a sample of 10 g of juveniles was taken from the same pool to determine the proximate chemical composition and fatty acid profile of the juvenile whole-body (see Section 2.3).

Water from a deep wheel was supplied in an open system (flow-throughout), and each tank had a water inlet (inflow 0.30 L min^−1^) and outlet (provided with a 250 µm mesh filter) and light aeration. Measures of the incoming water quality, ammonia, nitrites, hardness, and total suspended solids were performed once a week with a spectrophotometer HACH DR2800 (Hach Lange GMBH, Vigo, Spain). Dissolved oxygen in tanks was measured with a multi-meter HACH HQ30d (Hach Lange GMBH, Vigo, Spain). The mean values of water quality were pH = 7.5; hardness, 5.3 German degrees (calcium 32.8 mg L^−1^); total suspended solids, 34.0 mg L^−1^; dissolved oxygen ranged between 5.9 and 7.4 mg L^−1^; ammonia <0.10 mg L^−1^; and nitrites <0.010 mg L^−1^. Water temperature was recorded twice a day and was maintained at 25 ± 1 °C. A photoperiod of 16 h light: 8 h dark was maintained. Tanks were cleaned of feces and uneaten feed every two days. 

The experiment lasted for 90 days.

The Ethics Committee of the León University (Spain) approved all procedures used in the study. 

### 2.2. Diets and Feeding

Six diets (nearly isonitrogenous and isoenergetic) with different replacement levels of FO by a mixture of VO were formulated: 0% (control), 20%, 40%, 60%, 80%, or 100%. The mixture of vegetable oils was 30% linseed oil, 20% corn oil, and 50% olive oil. The ingredients were ground in a rotary BRABENDER mill (Brabender GmbH & Co. KG, Duisburg, Germany), mixed in a STEPHAN UMC5 mixer (Stephan Food Service Equipment, Hameln, Germany), and extruded using a stand-alone extruder BRABENDER KE19/25D (Brabender GmbH & Co. KG, Duisburg, Germany) at a temperature range between 100 °C and 110 °C. Pellets (1 mm diameter) were dried during 24 h at 30 °C, and afterwards, received a coating of cod liver oil and/or the mixture of VOs. Formulation of the practical diets is presented in Table 1.

### 2.3. Chemical Analysis of Diets and Fish

Samples of diets and juvenile samples, at the beginning and the end of the experiment, were stored at −30 °C. Juveniles were fasted for 14 h before sampling. Analyses were performed in duplicate by Analiza Calidad laboratory (Burgos, Spain) following Commission Regulation (EC) 152/2009. Moisture was determined by drying at 105 °C, crude protein according to the Kjeldahl method, crude lipid by extraction with light petroleum and further distillation, ash by calcination at 550 °C, and gross energy according to EU regulation 1169/2011. The content of carbohydrates was calculated by subtracting the content of moisture, protein, lipid, and ash from the wet weight.

Fatty acid profiles were determined by hydrophilic interaction chromatography (HPLC) using the AccQTag method from Waters (Milford, MA, USA).

Proximate composition and fatty acid profile of diets are presented in Table 2 and Table 3, respectively. 

### 2.4. Data Collection

Juvenile tench behavior was observed and registered after cleaning and feeding tasks.

TL and W of a sample of 15 fishes per tank (45 per treatment, 50% of total) were measured after 30 and 60 days of the experiment were measured following the procedure described in Section 2.1, with the aim to obtain information on the evolution of growth. Afterwards, juveniles were gently returned to their respective tanks. 

After 90 days, surviving fishes were anesthetized and observed one by one using a magnifying glass to detect externally visible deformities affecting spinal axis, operculum, mouth, and tail fin. Individual weight and the total length of all fish were measured, and total biomass per tank was calculated. In addition, the following indices were calculated:-Survival rate (%): (final number of juveniles × initial number of juveniles^−1^) × 100-Specific growth rate, SGR (% d^−1^) = 100 × ((ln final W − ln initial W) × days^−1^)).-Fulton’s coefficient or condition factor, K = 100 × (W × TL^−3^).-Biomass gain, BG (g) = final Biomass per tank − initial Biomass per tank. -Feed conversion ratio, FCR = total amount of feed supplied per tank × BG^−1^ [45].

### 2.5. Statistical Analysis

All treatments were replicated three times, and the experimental unit was a tank. Statistical analysis of growth performance and whole-body composition data was made using the SPSS16.0 computer program (SPSS, Chicago, IL, USA) by one-way analysis of variance (ANOVA) followed by, in the case of growth performance and whole-body composition, polynomial contrasts. Significant differences between means were estimated by the Tukey’s multiple range test. *p* < 0.05 was used for rejection of null hypothesis.

## 3. Results

### 3.1. Composition of Diets

There were no significant differences in proximate composition of experimental diets (Table 2). As can be seen in Table 3, compared to the control group, a significant reduction of myristic (14:0), pentadecanoic (15:0), palmitoleic (16:1), EPA, and DHA was observed in diets with 60% of FO replacement or higher. In the case of myristoleic acid (14:1), the reduction was significant in all diets that included partial or total FO replacement. Finally, 18:2-n6 (linoleic acid, LA) and 18:3-n3 (linolenic acid, ALA) content significantly increased from 60% FO substitution level.

From observations of juvenile tench behaviour, diets were equally ingested throughout the experiment, independently of the amount of vegetable oil blend included. Juvenile tench did not show abnormal behaviour.

### 3.2. Growth Performance

Table 4 shows total length and weight of juvenile tench after 30 and 60 days of the experiment. There were no significant differences between experimental diets for TL and W after 30 (mean value range: 31.37–33.14 mm and 0.44–0.57 g) and 60 days of the experiment (mean value range: 42.64–43.88 mm and 1.07–1.2 g). There were mean TL increases of 35.7% and 83% after 30 and 60 days, respectively. The mean W was tripled after 30 days, whereas at 60 days, was almost eight times higher.

In all cases, survival was 100%. Growth performance values and indices over 90 days are in Table 5. There were no significant differences on TL, W, SGR, FCR, K, and biomass gain (BG), showing the feasibility to full FO replacement without adverse effects on growth performance. No fish with externally visible deformities were observed.

### 3.3. Juvenile Tench Whole-Body Composition

Proximate composition and fatty acid profile in the whole-body at the beginning and end of the experiment are summarized in Table 6 and Table 7. Diets affected the composition of juveniles, so that lipid content was significantly lower in animals fed 100% of VO blend. Considering the initial composition, after 90 days, the whole-body content of lipids and fatty acids myristic (14:0), pentadecanoic (15:0), palmitoleic (16:1), eicosadienoic (20:2-n6), and gamolenic (18:3n-6) increased significantly, whereas content of eicosanoic (20:0), tetracosanoic (24:0), elaidic (18:1n-9), tetracosanoic (24:1), and araquidonic (20:4n-6) decreased. Moreover, a significant decrease (*p* < 0.05) of MUFA and a significant increase of PUFA were evidenced (Table 7).

After 90 days, the content of ALA (linolenic acid, 18:3n-3) in whole-body juvenile increased as FO substitution did, being significantly higher with respect to control diet from 40% FO replacement diets. No differences in ΣSFA, ΣMUFA, ΣPUFA, and Σn-3 and Σn-6 were found. 

Table 8 includes some indices commonly used to determine the nutritional and healthy value of fishes as food for humans. The ΣPUFA/ΣSFA and Σn-6/Σn-3 ratios tended to increase with dietary VO, whereas the EPA + DHA showed an opposite tendency.

## 4. Discussion

In natural habitats, the presence of planktonic crustaceans and other invertebrates in its digestive tract revealed that juvenile tench acts as a predator [46,47], and thus, vegetable oils are absent from its natural diet. For this reason, the inclusion of vegetable oils in the diet could have negative effects on palatability and, consequently, on feed intake. In our experiment, tench juveniles accepted all diets equally, showing that flavor and taste were acceptable, and, as Turchini et al. [13] suggested, the lipid fraction plays a minor role to determine the palatability of aquafeeds.

In some studies performed in other cyprinid species, such as black carp (*Mylopharyngodon piceus*) [32] and the minnow (Onchystoma macrolepis) [33] with rapeseed oil, and common carp (Cyprinus carpio) with unrefined peanut oil [34], sunflower oil, [35], or grapeseed oil [36], total replacement of FO did not adversely affect fish growth performance. The results of the present study also evidenced that total replacement of FO by VO was feasible without negative effects on growth parameters, leading us to consider that essential fatty acid (EFA) requirements were fully covered. It is important to consider that the requirements of EFA vary qualitatively and quantitatively between species, but also during the ontogeny of fish, with the early developmental stages being a critical period [14]. In this experiment, juvenile tench have a lower initial size and weight than those reported in other studies carried out in freshwater species, leading us to consider that the amount of EFA provided by the diets could be considered as a good approach to the minimum values required in this species.

Unlike marine fish, EFA requirements in freshwater species can generally be satisfied by the C:18 PUFA, because they are able to convert linoleic and linolenic acids to n-6 and n-3 long chain PUFAs, such as eicosapentaenoic (EPA, 20:5n-3) and docosahexaenoic (DHA, 22:6n-3) acids [48]. Considering this, the quantities of corn, linseed, and olive oil in the blend tested were estimated to obtain a similar or higher supply of LA an ALA than the cod liver oil, and, in fact, the content of both fatty acids increased as VO did (see Table 3). The control diet tested in this study has the same formulation that was tested by Sáez-Royuela et al. [43], but ALA content was lower (1.83 kg^−1^ and 4.60 g kg^−1^, respectively), whereas the content of LA was similar (7.2 and 6.7 g kg^−1^, respectively). The good growth performance obtained in both cases suggests that the ALA requirement for juvenile tench could be lower than the recommended level in freshwater fish [14], between 0.5 and 1% of diet dry weight.

Diet reflects final fish composition, and it could affect its nutritional properties and quality as human food [10]. With respect to whole-body proximate composition, a significant decrease of lipid content was evidenced when tench juvenile were fed on 100%VO compared to those fed the control diet. This disagrees with what has been reported in other freshwater species, such as Nile tilapia [30], beluga sturgeon [25], rainbow trout [15], and mandarin fish [27], where the inclusion of VO in the diet entails an increase of lipids in the whole-body and/or other tissues. In a study with juvenile golden pompano (*Trachinotus ovatus*), Guo et al. [49] found a strong association between body lipid content and dietary ALA/LA, so that the lowest body lipid content coincided with the highest ALA/LA ratio. We evidenced the same relationship in our experiment, where the high lipid deposition corresponded to juveniles fed on the control diet (ALA/LA = 0.25), and the low lipid deposition to juveniles fed on total FO replacement (ALA/LA = 0.37).

The long n-3 LC PUFA are abundant in fishes and, because they play important roles in human growth and the maintenance of health through preventing chronic cardiovascular diseases, diabetes, cancer, and age-related degenerative diseases, the consumption of fish products is highly recommended [50]. A negative effect derived from FO replacement by VO is the reduction of two long chain polyunsaturated fatty acids, EPA and DHA, in whole-body and/or fillets. This fact has been extensively documented in many recent studies, both in marine species, such as gilthead seabream (*Sparus aurata*) [51,52], European Seabass (*Dicentrarchus labrax*), [53], shortfin corvina (Cynoscion parvipinnis) [54], hybrid grouper (*Epinephelus fuscoguttatus* ♀ × *E. lanceolatus* ♂) [55], yellow croaker (*Larimichthys crocea*) [56], meagre (*Argyrosomus regius*) [57], and yellowtail (*Seriola dumerili*) [58], and also in freshwater species, such as silver barb [26], black carp [32], largemouth bass [24], Manchurian trout [21], Arctic and brook charr [20], and rainbow trout [21,59]. In this experiment, although EPA and DHA content in diets decreased significantly as VO increased, no differences in n-3 LC PUFA of whole-body juvenile tench were found. This is consistent with the results of Garrido et al. [60] about the high capacity of tench to biosynthesize n-3 LC PUFA from ALA. 

Similar to other studies performed freshwater species [19,20,21,26,27,33,59], the inclusion of dietary vegetable oils resulted in significant increase of ALA in the juvenile tench whole-body composition. Dietary intake of ALA is inversely correlated with cardiovascular disease and cancer risk in humans [61], and thus, the blend of VO used to replace FO seems to have a positive effect on the nutritional quality of tench. 

Between indices used for evaluating the nutritional value of human foods, Σ PUFA/Σ SFA is one of the most commonly used to evaluate the nutritional value and healthiness of foods. In the case of fishes, a wide range of values, between 0.50 and 1.62, has been reported depending on fish species [62], the higher being the better. In this experiment, this ratio varied between 0.70 and 0.86, corresponding to control and total FO replacement, respectively. Our values were lower than the reported in flesh of commercial sized tench, cultured under semi-intensive (average weight 680 g, Σ PUFA/Σ SFA: 1.10) and extensive conditions (average weight 210 g, Σ PUFA/Σ SFA: 1.32) [63]. Nevertheless, this index was over 0.40, the minimum limit for foods regarded as undesirable for human health, due to their potential effect of increasing blood cholesterol levels [64,65]. 

Another index used to evaluate the nutritional and healthy value of fish is the EPA + DHA value [62]. In tench cultured under semi-intensive conditions, this index was 3.19 g kg^−1^ [63], whereas the absolute values in our study were higher, between 7.88 g kg^−1^ in 100%VO diet, and 13.2 g kg^−1^ in control diet. The higher values obtained in our study could be attributed not only to the different size of fishes, but also to feeding conditions. In this sense, Linhartovà et al. [63] found a lower EPA + DHA and lipid content in tench reared under extensive conditions, with only natural food sources, than those cultured under semi-intensive conditions. In our study, juvenile tench only received extruded diets, leading to a higher lipid deposition than that reported in the above-mentioned study [63], and consequently, higher EPA + DHA content.

Finally, the Σ n-6/Σ n-3 ratio is a useful indicator not only for comparing the relative nutritional value of fish of different species, but also to establish a healthy diet in humans, where the recommended optimal ratio of omega-6/omega-3 varies from 1/1 to 4/1, depending on the disease under consideration [66]. Considering data reported by Wereńska et al. [67], this ratio varies significantly among fish species, being higher in freshwater species. The values obtained in our study between 0.43 and 0.62 did not differ to a great extent to the reported in animals from semi-intensive and extensive cultures: 0.75 and 0.47, respectively [63]. 

It must be taken into account that comparisons between nutritional indices values for tench available in the scientific literature and our results should be interpreted with carefulness, because we used small-sized animals, the fatty acid profile was analyzed in the whole-body, and tench did not receive natural food. Although the information provided could be considered as an initial approach, further research on the effects of VO diet on the nutritional quality in tench reared to commercial size should be carried out. 

## 5. Conclusions

A total replacement of FO by a blend of VO did not affect the survival and growth performance of juvenile tench. Lipid content in the whole-body was significantly lower in the 100% FO replacement diet compared to control diet. There were no significant differences in the whole-body content of n-3 LC PUFA found, whereas an increase of ALA was evidenced in tench fed from 40%VO diets. Other nutritional indexes, Σ PUFA/Σ SFA, DHA + EPA, or Σn-6/Σn-3, can be considered nutritionally adequate and healthy for human feeding. Further research on the effects of VO diet on the nutritional quality in tench reared to commercial size should be performed.

## Figures and Tables

**Table 1 animals-12-01113-t001:** Formulation of the experimental diets (g kg^−1^ diet) with different levels of replacement of cod liver oil by a blend of vegetable oils (30% linseed oil, 20% corn oil, and 50% olive oil).

Ingredients (g kg^−1^)	FO Replacement (%)					
	0	20	40	60	80	100
Fish meal ^1^	645	645	645	645	645	645
Corn meal ^2^	166	166	166	166	166	166
Dried *Artemia* cysts ^3^	100	100	100	100	100	100
Carboxymethyl *cellulose* ^4^	30	30	30	30	30	30
Mixture of vegetable oils	0	4	8	12	16	20
Cod liver oil ^5^	20	16	12	8	4	0
L-ascorbyl-2-monophosphate-Na ^6^	5	5	5	5	5	5
*Dicalcium phosphate* ^6^	10	10	10	10	10	10
Choline chloride ^6^	3	3	3	3	3	3
Soy lecithin ^7^	10	10	10	10	10	10
Sodium chloride ^8^	1	1	1	1	1	1
Mineral and Vitamin premix ^9^	10	10	10	10	10	10

^1^ Skretting España S.A., Ctra. de la Estación s/n 09,620 Cojóbar. Burgos. España; ^2^ Adpan Europa S.L., ES-33186 El Berrón. Siero. Asturias. Spain; ^3^ INVE Aquaculture Nutrition. Hoogyeld 91. Dendermonde. Belgium; ^4^ Helm Iberica S.A., ES-28108 Alcobendas. Madrid. Spain; ^5^ Acofarma distribution S.A., ES-08223 Terrassa. Barcelona. Spain; ^6^ Cargill., ES-28720 Colmenar Viejo. Madrid. Spain; ^7^ Biover N.V., Monnikenwerve 109. B-8000 Brugge. Belgium; ^8^ Unión Salinera de España S.A., ES-28001 Madrid. Spain; ^9^ Provides mg kg-1 premix: inositol, 50,000; thiamin, 500; riboflavin, 800; niacin, 5000; pyridoxine, 1500; pantothenic acid, 5000; biotin, 150; folic acid, 3500; cyanocobalamin, 5; retinol, 2400; α-tocopherol, 30,000; cholecalciferol, 6.25; naphthoquinone, 5000; butylated hydroxytoluene, 1500; MgSO_4_-7H_2_O, 300,000; ZnSO_4_-7H_2_O, 11,000; MnSO_4_-H_2_O, 4000; CuSO_4_-5H_2_O, 1180; CoSO_4_, 26; FeSO_4_-7H_2_O, 77,400; KI, 340; Na_2_SeO_3_, 68.

**Table 2 animals-12-01113-t002:** Proximate composition (g kg^−1^ diet, wet basis) of the experimental diets with different levels of replacement of cod liver oil (FO) by a blend of vegetable oils.

FO Replacement (%)						
Proximate Composition (g kg^−1^)	0	20	40	60	80	100
Moisture	72.0 ± 2.0	71 ± 1.5	70.1 ± 3.2	74.1 ± 3.5	69.8 ± 2.8	71.6 ± 2.3
Crude protein	482 ± 12.0	482 ± 11.0	471 ± 12	481 ± 13.0	478 ± 10.0	479 ± 9.0
Crude lipid	111.0 ± 3.4	113.5 ± 2.6	117.8 ± 3.1	114.0 ± 2.9	110.8 ± 2.8	112.4 ± 3.0
Carbohydrates	205.0 ± 4.2	203.3 ± 3.0	203.0 ± 3.2	204.3 ± 3.9	211.4 ± 4.4	200.0 ± 3.6
Ash	130.0 ± 1.9	130.2 ± 2.0	128.1 ± 1.8	126.6 ± 1.7	130 ± 1.8	137.0 ± 1.7
Gross energy (MJ kg^−1^)	15.70 ± 0.7	15.73 ± 0.8	15.87 ± 0.6	15.72 ± 0.7	15.71 ± 0.7	15.6 ± 0.6

Values are presented as mean ± standard error of the mean (SEM).

**Table 3 animals-12-01113-t003:** Fatty acid profile (% of total lipid content) of the practical diets with different levels of replacement of cod liver oil by a blend of vegetable oils.

FO Replacement (%)						
Fatty Acids	0	20	40	60	80	100
14:0	5.11 ± 0.11 ^a^	5.48 ± 0.26 ^a^	5.48 ± 0.20 ^a^	3.16 ± 0.16 ^b^	3.05 ± 0.10 ^b^	3.00 ± 0.10 ^b^
15:0	0.47 ± 0.02 ^a^	0.53 ± 0.02 ^a^	0.52 ± 0.02 ^ab^	0.35 ± 0.03 ^b^	0.38 ± 0.02 ^b^	0.39 ± 0.02 ^b^
16:0	18.73 ± 0.50	18.53 ± 0.04	18.41 ± 0.23	18.35 ± 0.35	18.00 ± 0.20	18.12 ± 0.19
17:0	0.64 ± 0.02	0.54 ± 0.02	0.53 ± 0.02	0.49 ± 0.06	0.48 ± 0.04	0.48 ± 0.03
18:0	3.95 ± 0.10	4.56 ± 0.31	4.44 ± 0.32	4.25 ± 0.28	4.99 ± 0.33	4.95 ± 0.34
20:0	1.36 ± 0.06	1.61 ± 0.20	1.59 ± 0.18	1.26 ± 0.15	1.14 ± 0.21	1.04 ± 0.19
24:0	0.32 ± 0.02	0.32 ± 0.01	0.27 ± 0.02	0.24 ± 0.01	0.24 ± 0.02	0.23 ± 0.01
14:1	0.26 ± 0.03	0	0	0	0	0
16:1	8.5 ± 0.20 ^a^	8.10 ± 0.10 ^ab^	7.98 ± 0.06 ^ab^	7.32 ± 0.13 ^bc^	6.53 ± 0.12 ^c^	5.66 ± 0.14 ^d^
17:1	0.63 ± 0.02	0.61 ± 0.05	0.60 ± 0.04	0.56 ± 0.02	0.57 ± 0.03	0.53 ± 0.02
18:1n-9	23.37 ± 0.47	24.05 ± 1.50	24.73 ± 1.30	25.79 ± 1.40	26.20 ± 1.30	26.50 ± 1.10
20:1	2.44 ± 0.30	2.49 ± 0.21	2.55 ± 0.30	3.00 ± 0.27	3.10 ± 0.29	3.60 ± 0.31
22:1n-9	1.85 ± 0.02	1.91 ± 0.22	1.87 ± 0.18	1.51 ± 0.15	1.52 ± 0.14	1.43 ± 0.13
24:1	1.01 ± 0.02	1.02 ± 0.07	1.10 ± 0.09	1.11 ± 0.10	1.10 ± 0.12	0.99 ± 0.08
18:2-n6	6.50 ± 0.30 ^a^	7.40 ± 0.50 ^ab^	7.77 ± 0.42 ^abc^	7.66 ± 0.18 ^bcd^	9.51 ± 0.55 ^cd^	11.62 ± 0.67 ^d^
18:3n-6	<0.05	<0.05	0.07 ± 0.06	0.12 ± 0.06	0.26 ± 0.06	0.31 ± 0.06
18:3n-3	1.65 ± 0.04 ^a^	2.31 ± 0.13 ^ab^	2.28 ± 0.12 ^ab^	2.82 ± 0.14 ^bc^	3.34 ± 0.19 ^c^	4.28 ± 0.21 ^d^
20:4n-6	0.78 ± 0.03	0.98 ± 0.09	1.05 ± 0.08	0.85 ± 0.09	0.75 ± 0.06	0.78 ± 0.05
20:5n-3	9.48 ± 0.20 ^a^	8.50 ± 0.23 ^ab^	8.34 ± 0.24 ^abc^	8.20 ± 0.21 ^bc^	7.95 ± 0.19 ^bc^	7.24 ± 0.18 ^c^
22:6n-3	7.67 ± 0.37 ^a^	6.77 ± 0.29 ^ab^	6.65 ± 0.26 ^ab^	6.47 ± 0.24 ^abc^	5.87 ± 0.26 ^bc^	5.03 ± 0.21 ^c^
Σ SFA ^1^	30.58 ± 0.79	31.57 ± 1.10	31.24 ± 1.30	28.10 ± 0.90	28.28 ± 0.85	28.21 ± 0.79
Σ MUFA ^2^	38.05 ± 1.16	38.08 ± 1.98	38.83 ± 1.50	39.29 ± 1.53	39.02 ± 1.48	38.71 ± 1.62
Σ PUFA ^3^	26.08 ± 0.94	25.96 ± 1.03	26.69 ± 1.02	27.96 ± 1.22	29.00 ± 1.09	29.26 ± 1.09
ALA/LA ^4^	0.25 ± 0.006 ^a^	0.31 ± 0.003 ^ab^	0.29 ± 0.05 ^ab^	0.32 ± 0.007 ^ab^	0.32 ± 0.005 ^ab^	0.37 ± 0.004 ^b^

Values are presented as mean and standard error of the mean (SEM). Means in the same row with different superscripts are significantly different. Some minor fatty acids (<0.05%) are not shown. ^1^ Total saturated fatty acids. ^2^ Total monounsaturated fatty acids. ^3^ Total polyunsaturated fatty acids. ^4^ Linolenic acid/Linoeic acid.

**Table 4 animals-12-01113-t004:** Growth performance of juvenile tench fed experimental diets with different levels of cod liver oil replacement by a blend of vegetable oils over 30 and 60 days.

Days			FO Replacement							Polynomial Contrasts		
		0	20	40	60	80	100	SEM	ANOVA	Linear	Quadratic	Cubic
**30**	TL ^1^ (mm)	33.15	31.84	31.82	32.82	33.33	31.37	0.334	0.46	0.62	0.88	0.05
	W ^2^ (g)	0.57	0.44	0.45	0.47	0.50	0.42	0.019	0.30	0.19	0.38	0.06
**60**	TL ^1^ (mm)	43.26	44.23	42.60	43.95	43.93	44.92	0.371	0.63	0.30	0.47	0.70
	W ^2^ (g)	1.18	1.12	1.08	1.16	1.19	1.29	0.033	0.59	0.240	0.17	0.83

Values are presented as mean and pooled standard error of the mean (SEM). ^1^ Total length; ^2^ Weight.

**Table 5 animals-12-01113-t005:** Growth performance of juvenile tench fed experimental diets with different replacement of cod liver oil by a blend of vegetable oils over 90 days.

Growth Performance	FO Replacement (%)								Polynomial Contrasts		
	0	20	40	60	80	100	SEM	ANOVA	Linear	Quadratic	Cubic
TL ^1^ (mm)	50.09	51.34	49.98	50.76	50.84	51.24	0.285	0.71	0.43	0.83	0.55
W ^2^ (g)	1.92	1.91	1.86	1.82	1.87	1.94	0.361	0.96	0.93	0.43	0.67
SGR ^3^ (% day^−1^)	2.65	2.70	2.64	2.69	2.74	2.74	0.021	0.71	0.20	0.74	0.99
K ^4^	1.35	1.31	1.35	1.31	1.36	1.34	0.068	0.13	0.51	0.24	0.38
FCR ^5^	1.29	1.28	1.27	1.32	1.30	1.28	0.034	0.20	0.04	0.96	0.62
BG ^6^ (g)	53.83	54.36	53.02	50.21	52.21	53.91	1.074	0.94	0.69	0.45	0.64

Values are presented as mean and pooled standard error of the mean (SEM). ^1^ Total length; ^2^ Weight; ^3^ Specific growth rate = 100 × ((ln final body weight − ln initial body weight) × days)^−1^); ^4^ Condition factor = 100 × (body weight × ((body length^3^) ^−1^); ^5^ Feed conversion ratio = total amount of feed supplied per tank × BG^−1^; ^6^ Biomass gain = final Biomass per tank − initial Biomass per tank.

**Table 6 animals-12-01113-t006:** Proximate composition (g kg^−1^, wet basis) of the whole-body of juvenile tench fed experimental diets with different levels replacement of cod liver oil by a blend of vegetable oils.

Proximate Composition			FO Replacement (%)							Polynomial Contrasts		
	Initial	0	20	40	60	80	100	SEM	ANOVA	Linear	Quadratic	Cubic
Moisture	799.9	753.5	763.4	756.4	770.2	772.4	752.8	7.64	0.77	0.16	0.97	0.61
Protein	137.0	140.0	152.5	153.8	142.3	150.8	156.7	2.61	0.29	0.13	0.19	0.006
Lipid	31.3 ^a^	77.0 ^b^	75.6 ^b^	76.5 ^b^	67.1 ^bc^	69.6 ^bc^	64.2 ^c^	4.32	<0.001	<0.001	0.01	0.38
Ash	24.9	27.4	23.4	25.1	22.0	20.4	22.4	0.70	0.07	0.50	0.009	0.24

Values are presented as mean and pooled standard error of the mean (SEM). Means in the same row with different superscripts are significantly different (*p* < 0.05).

**Table 7 animals-12-01113-t007:** Fatty acid profile (% of lipids) of the whole-body of juvenile tench fed experimental diets with different levels replacement of cod liver oil by a blend of vegetable oils.

Fatty Acids			FO Replacement (%)							Polynomial Contrasts		
	Initial	0	20	40	60	80	100	SEM	ANOVA	Linear	Quadratic	Cubic
14:0	1.53 ^a^	3.42 ^b^	3.00 ^b^	2.92 ^b^	3.47 ^b^	3.15 ^b^	2.92 ^b^	0.17	0.001	<0.001	0.83	0.96
15:0	<0.05 ^a^	0.42 ^b^	0.37 ^b^	0.38 ^b^	0.44 ^b^	0.38 ^b^	0.37 ^b^	0.04	0.001	<0.001	0.70	0.86
16:0	15.38	21.72	21.08	20.28	21.30	21.28	20.46	0.76	0.34	0.03	0.97	0.89
17:0	0.35	0.32	0.30	0.30	0.34	0.29	0.29	0.01	0.42	0.13	0.40	0.64
18:0	3.83	3.21	3.10	3.35	3.05	3.18	3.12	0.09	0.37	0.04	0.63	0.91
20:0	0.52 ^a^	0.20 ^b^	0.18 ^b^	0.19 ^b^	0.18 ^b^	0.18 ^b^	0.19 ^b^	0.03	<0.001	<0.001	0.17	0.67
24:0	1.26 ^a^	<0.05 ^b^	0.30 ^c^	0.27 ^c^	0.33 ^c^	0.32 ^c^	0.25 ^c^	0.10	<0.001	<0.001	0.04	0.001
14:1	<0.05 ^a^	<0.05 ^a^	<0.05 ^a^	0.29 ^bc^	0.33 ^b^	0.30 ^bc^	0.26 ^c^	0.04	<0.001	<0.001	<0.001	<0.001
16:1	5.59 ^a^	11.94 ^b^	11.12 ^b^	11.33 ^b^	11.00 ^b^	10.87 ^b^	10.41 ^b^	0.60	0.004	<0.001	0.78	0.77
17:1	0.69	0.70	0.65	0.64	0.69	0.62	0.59	0.02	0.90	0.66	0.35	0.88
18:1n-9	48.52 ^a^	31.46 ^b^	34.10 ^b^	33.51 ^b^	33.3 ^b^	33.24 ^b^	33.10 ^b^	1.57	0.001	<0.001	0.98	0.90
20:1	1.26 ^a^	2.44 ^ab^	2.49 ^ab^	2.55 ^ab^	3.00 ^b^	3.10 ^b^	3.60 ^b^	0.31	0.008	0.001	0.004	0.30
22:1n-9	0.44 ^a^	0.75 ^b^	0.62 ^ab^	0.66 ^ab^	0.76 ^b^	0.66 ^ab^	0.60 ^ab^	0.03	0.047	0.004	0.46	0.71
24:1	0.77 ^a^	<0.05 ^b^	<0.05 ^b^	<0.05 ^b^	<0.05 ^b^	<0.05 ^b^	<0.05 ^b^	0.07	<0.001	<0.001	0.003	0.89
18:2n-6	7.11	5.36	6.95	6.94	6.82	7.37	7.71	0.30	0.60	0.66	0.11	0.64
20:2n-6	<0.05 ^a^	0.19 ^b^	0.20 ^b^	0.16 ^b^	0.18 ^b^	0.21 ^b^	0.18 ^b^	0.02	0.001	<0.001	0.40	0.61
18:3n-6	<0.05 ^a^	0.21 ^b^	0.19 ^b^	0.19 ^b^	0.21 ^b^	0.20 ^b^	0.19 ^b^	0.02	0.001	<0.001	0.55	0.92
18:3n-3	1.38 ^a^	1.37 ^a^	2.48 ^ab^	2.92 ^b^	2.74 ^b^	2.93 ^b^	3.33 ^b^	0.21	0.002	0.002	<0.001	0.06
20:3n-6	<0.05 ^a^	0.28 ^b^	0.32 ^b^	0.30 ^b^	0.29 ^b^	0.30 ^b^	0.31 ^b^	0.03	<0.001	<0.001	0.06	0.86
20:4n-6	1.07 ^a^	0.15 ^b^	0.17 ^b^	0.19 ^b^	0.16 ^b^	0.16 ^b^	0.19 ^b^	0.09	<0.001	<0.001	0.06	0.84
20:5n-3	3.25	5.34	4.41	4.58	5.23	4.46	4.30	0.19	0.14	0.01	0.25	0.46
22:6n-3	6.53	7.64	6.27	6.52	6.57	6.28	6.25	0.19	0.56	0.81	0.16	0.38
ΣSFA ^1^	22.87	29.29	28.33	27,69	29.11	28.78	27.60	0.69	0.06	0.003	0.41	0.35
ΣMUFA ^2^	57.27 ^a^	47.29 ^b^	48.99 ^b^	49.05 ^b^	49.08 ^b^	48.79 ^b^	48.56 ^b^	1.00	0.01	<0.001	0.61	0.60
ΣPUFA ^3^	12.81 ^a^	20.54 ^b^	20.99 ^b^	21.80 ^b^	21.50 ^b^	21.91 ^b^	22.46 ^b^	0.89	0.001	<0.001	0.04	0.95
Σn-6	8.18	6.19	7.83	7.78	7.66	8.24	8.58	0.26	0.27	0.42	0.03	0.38
Σn-3	14.41	14.35	13.16	14.02	13.84	13.67	13.88	0.27	0.95	0.56	0.84	0.72

Values are presented as mean and pooled standard error of the mean (SEM). Means in the same row with different superscripts are significantly different (*p* < 0.05). Some minor fatty acids (<0.05%) are not shown. ^1^ Total saturated fatty acids; ^2^ Total monounsaturated fatty acids; ^3^ Total polyunsaturated fatty acids.

**Table 8 animals-12-01113-t008:** Nutritional indices of the whole-body of juvenile tench fed experimental diets with different levels re-placement of cod liver oil (FO) by a blend of vegetable after 90 days.

Nutritional Indices			FO Replacement (%)							Polynomial Contrasts	
	0	20	40	60	80	100	SEM	ANOVA	Linear	Quadratic	Cubic
ΣPUFA/ΣSFA ^1^	0.70	0.74	0.79	0.74	0.76	0.86	0.01	0.68	0.38	0.37	0.52
EPA + DHA ^2^	9.99	8.07	8.49	7.45	7.47	6.67	0.36	0.44	0.15	0.38	0.32
Σn-6/Σn-3	0.43	0.60	0.56	0.59	0.61	0.62	0.03	0.59	0.18	0.51	0.47

Values are presented as mean and pooled standard error of the mean (SEM); ^1^ Σ polyunsaturated fatty acids/Σ saturated fatty acids (g kg^−1^ wet weight); ^2^ Eicosapentaenoic acid + docosahexaenoic acid (g kg^−1^ wet weight).

## Data Availability

Data are available on corresponding author’s request.

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
