# Peer review of "Fish Oil Replacement by a Blend of Vegetable Oils in Diets for Juvenile Tench (Tinca tinca Linnaeus, 1758): Effects on Growth Performance and Whole-Body Composition"

_animals, 2022, doi:10.3390/ani12091113_

Round 1
Reviewer 1 Report
1- Correct “vegetal oils” to “vegetable oils” or “plant oils” in title and elsewhere in the text.
2- Lines 11-12: Correct the sentence in these lines to “Replacement of FO with VO at all levels led to the significant reduction of whole-body lipid content”.
3- Line 15: Change “whole-body juveniles” to “fish whole-body”.
4- Lines 15-16: The authors have conducted the experiment with Tench larvae, and the final mean weight of fish at the end of experiment was ~2 g, so what is the point of discussing about the nutritional value indices?
5- Line 26: Remove “daily increment in total length”.
6- Line 26: Correct “food conversion ratio” to “feed conversion ratio”.
6- Line 27-28: Correct the sentence in these lines to “Compared to the control group, lipid whole-body lipid content decreased significantly in 100%VO group.”
7- Lines 29 and 30: Remove “juveniles” and “juvenile’ after “whole-body”.
8- Line 32: Correct “tend to increase with dietary VO” to “tended to increase with increasing dietary VO content”.
9- Line 34: Remove “nutritional”.
10- Lines 40-42: Remove “in particular SDGs 1 (end poverty), 2 (end hunger), 5 (gender), 6 (water), 8 (growth, employment, decent work), 12 (production and consumption), 13 (climate change), 14 (marine resources & ecosystems) and 15 (biodiversity)”.
11- Line 45: Change “feedstuffs” to “aquafeeds”.
12- Lines 50-51: Correct “with a global use in 2019 of 78% and 68% of FM and FO, respectively” to “with a global use of 78% and 68% of FM and FO, respectively, in 2019”.
13- Line 55: Correct “long chain polyunsaturated n3 fatty acids (LC-PUFA n3) for finfishes, such as eicosapentaenoic acid (EPA) and docosahexaenoic acid (DHA)” to “n-3 long chain polyunsaturated fatty acids (n-3 LC-PUFA) such as eicosapentaenoic acid (EPA) and docosahexaenoic acid (DHA) for finfish”.
14- Line 71 and elsewhere: Correct “LC-PUFA n3” to “n-3 LC-PUFA”.
15- Line 81 and 83: Correct “healthy properties” to “healthiness properties”.
16- Lines 86-87: Correct “Although is considered one of the most promising species for the development of inland culture, production is based…” to “Although it is considered as one of the most promising species for diversification of inland aquaculture, its production is based…”.
17- Line 93: Correct the format of references in this line.
18- Line 97: Correct “know” to “evaluate”.
19- Change “practical diets” to “experimental diets” in caption of all tables.
20- Table 1: Please present the dietary formulation as dry matter basis not wet basis.
21- Line177: Correct the formula for calculation of survival rate.
22- Line 179: Remove this parameter.
23- Lines 190-193: Provide the statements in these lines in the materials and methods section.
24- Lines 195-196: Correct “Comparing with the control diet” to “Compared to the control group”.
25- Lines 211-212: Instead of talking about observation of fish feed ingestion, you need to present fish feed intake data.
26- Line 241: Correct “between of” to “for”.
27- Line 225: Correct “fully” to “full”.
28- Lines 280-282 and elsewhere: Provide the scientific name of the discussed species.
29- Rephrase the sentences in lines 288-291, 298-299, 303-305.
30- Line 302: Use “level” after “recommended”.
31- Line 313: Correct “correspond” to “corresponded”.
32- Line 332: Remove “the reported in”.
33- Line 334-336: Rephrase the sentence in these lines.
34- Line 341: Remove “diet” after “control”.
35- Line 344: Remove “in the Czech Republic”.
36- Line 358: Correct “an useful” to “a useful”.
37- Line 360: Correct “optimal ratio of omega-6/omega-3 recommended” to “recommended optimal ratio of omega-6/omega-3”.
38- Correct the ending part of the sentence in lines 369-371.
Author Response
First of all, we want to thank your corrections that were fully considered
Also, some major changes proposed have been taken into account: ITL index was removed and statements in lines 190-193 moved to materials and methods section.
Also we have rephrased the sentence in lines 288-291, 298-299, 303-305 and 334-346.
Lines 15-16: About the question to the point to discussing about nutritional value indices:
We feel that, although the weight of fishes at the end of the experiment is low, is interesting highlight that VO blend did not have negative effects of nutritional quality of tench. However, as we stated in the discussion, we are concious that further research is needed to know if a similar response will occur with commercial size tench.
Lines 211-212: About the fish feed ingestion:
The small size of tench makes imposible separate faeces from remainig feed and, thus, data of feed intake are not available. Nevertheless, in most of the observations made before diet supply, leftover feed were not clearly evidenced. Also, independently of the diet provided, almost immediate intake was observed leading us to hyphotesize that palatabilty was unaffected by the inclusión of the VO blend.
Sentences in lines 288-291, 298-299, 303-305 and 334-336 were rephrased.
Reviewer 2 Report
Concerning the manuscript entitled Fish Oil Replacement by a Blend of Vegetal Oils in Diets for Juvenile Tench (Tinca Tinca L.): Effects on Growth Performance and Whole-Body Composition. The manuscript is interesting, but I have some comments that should be considered for improving the manuscript.
L22. Please define that it is cod liver oil
L25-26. The abbreviations should be in parentheses
In general, the abstract needs revision.
L99. Hypothesis is missing, please add it
L103-105. The number of replicates per treatment should be added. If I correct, there are only 3 replicates per treatment and 6 groups. This is small number of replicates?
L124. Please provide ethical approval number
L166. Please separate between data collection and statistical analysis. Statistical analysis should be in a separate heading
L169. Please rephrase
L187, please correct Tuckeys’s
Tables 2 and 3 should be in the material and methods section please move them upward
Results
Please add subheadings to be easier for the readers
L207. It seems that the data are presented as means ± SE, please revise or confirm
L221. Please revise
L239. Revise
Why some data are presented as means ± SEM and others ± SE, please be consistent
Table 6, 7, and 8. Why the authors did not perform orthogonal polynomial contrasts for these data. It is important for the reader to show the effect of the incremental levels of vegetable oil on these studied parameters.
Why these data are presented as means ± SE? please be consistent throughout the manuscript.
L332, please rephrase
Author Response
First of all, we want to thank your comments and suggestions to improve our manuscript. Changes proposed in lines 21, 25-26, 166, 187, 231,239 were considered.
Following your advice Tables 2 and 3 were moved to Material and Methods section.
The editor ask to us for the ethical approval document and we already send to her. As it is stated in the text, the procedures were approved by the Ethics Committee of the University of León but unfourtunately does not have a number reference. For this reason it was not included in the text.
A separate heading was included for Statistical analysis.
Line 166 was rephrased
About the number of replicates, three replicates (tank=replicate) is a common and suitable number in this type of studies.
We have added subheadings in Results section
Tables 6, 7 and 8 were modified to include ortogonal polynomial constrasts.
Line 332 was rephrased
Reviewer 3 Report
The manuscript of María Sáez-Royuela and colleagues, focus on a very current topic, as the trials to substitute fish oil or fish meal with alternative sources, more cheapest and environmental sustainable, whit the double scope to improve gaining for aquaculture companies and preserve the marine natural stocks. As highlighted by bibliography, this is not the first contribution of this research group on the field, they are hence expert and skilled in this topic, and the manuscript construction confirm this. The study have an interesting value from a results point of view, but its soundness is in my opinion not really hight, due to several weakness and point to developing in future studies, that are needed for better assessing the results of this study, that maybe could be more comprehensive of other analysis and experiments.
By the way, focusing on the content of this document, I reported as follow my comments and suggestion on this manuscript, trying to improve it for a further publication, after addressing some major and minor point that I resume below.
Title
Title is informative and synthetic, but the specific name of the study model must be put correctly in italics and with the reference, as Tinca tinca (Linnaeus, 1758). Please take care of this suggestion also for the first time in which the species is mentioned in the manuscript sections.
Abstract and Simple summary needs adjustments based on the successive comments of the sections content.
Keywords
All the used keywords were already reported in Title. Please substitute it with some related different ones, in order to give more soundness to your manuscript during the web search phases.
Introduction
This section resumes correctly the topics treated in the manuscript, but needs more focus on the three different vegetables oil used in this study as trial to substitute fish oil. Pleas add some relate sentences with adequate references on teleost nutrition or related marine organisms.
Material and Methods
2.1 Please add as material the specifics of length-weight measure instruments.
Information about experimental location were totally missed. Where the experimental procedures have been carried out?
Experimental design its unclear and reduced to a sentence, maybe a summary table, also as supplementary material, should be inserted.
Line 117: what do the Authors mean with "Artesian water". Please give more accurate information related to the medium utilized in the study.
Results
Line 211: the present form of the manuscript show that only observation were used to evaluate the palatability of administered diets, this is a weak method. This kind of evaluation needs the daily collection of unconsumed feed from the tank bottoms, and an estimation by counting or weighting of the amount, to subtract it from the total of food ingested in subsequent calculations of the indices.
Lines 214-216: in this sentence the Authors declared "
There were no significant differences between of TL and W after 30 (mean value 214 range: 31.37-33.14 mm and 0.44-0.57 g) and 60 days of experiment (mean value range: 215 42.64-43.88 mm and 1.07-1.2 g)", but to which experimental diets is referring this sentence? Please clarify this unclear sentence.
Why values at 90 days were not considered together with the 30 days and 60 days ones? I don't understand this choice, also in realization of two separate Tables, the 4 and 5, for this similar data that were to merged and compared in a single table in my opinion.
Discussion
Lines 277-279: did the Authors evaluate that different flavor ingredients could not represent a deterrent for fish, but may lead to different taste of the final commercial products? (these fish were farmed for commercial purposes, as human consumption I guess).
Lines 319-330: Despite from the Author evaluation, the decreasing of EPA-DHA contents seems to be non representing a problem, would be useful if the Authors could purpose some ideas to avoid this problem with some insights for future studies.
Lines 332-365: despite the admission of the data weakness in lines 366-371, this period is affected by a big weakness of this study, the evaluation of experimental diets effect on lipids content based on a trial of the first 90 days of life cycle of the fish. This can't lead to a real key result for the field, and should be better highlighted in the manuscript.
Line 368: in this line were reported the second big weakness of this study, in my opinion. Why the experimental evaluations were based, from experimental design, on the whole body and not on the edible part? The manuscript sense assumes different significance on these regards in my opinion.
Conclusion
Please, add the last part of Discussion section in Conclusion, to better highlight the weakness and limitations of this study.
References
Please double check the references list and the referencing in main text, because there are some non uniforme styles.
Have a nice work.
Best regards
The Reviewer
Author Response
First of all, we want to thank your comments and suggestions to improve our manuscript.
Title
The complete scientific name of tench was included in the Title and also for the first time in which the species is mentioned in the text.
Keywords
Following your advice we have changed some Keywords.
Introduction
The two reviews cited (references10 and 11) give many references of the use of the three oils. In addition, at the of introduction we give an explanation for the election of vegetable oils used in the blend.
Material and Methods
We added the information of length-weight measure instruments and experimental location.
Experimental design: We have include an brief explanation of juvenile distribution and replication.
Water comes from a deep wheel used exclusively for the aquaculture facilities located in the University of León.
Results
The small size of tench makes imposible separate faeces from remainig feed and, thus, data of feed intake are not available. Nevertheless, in most of the observations made before diet supply, leftover feed were not clearly evidenced. Also, independently of the diet provided, almost immediate intake was observed leading us to hyphotesize that palatabilty was unaffected by the inclusión of the VO blend.
Lines 214-216: There were not differences between experimental diets for TL and W at 30 and 60 dyas of experiment.
The reason to not consider together 90-day growth performance values with 30 and 60-day values is that some growth indices can not be properly calculated in the two first sampling dates.
Discussion
Lines 277-279: The rapid ingestion of all experimental diets led us to consider that the blend of VO has not negative effects on palatabilty. However, as you suggest further research must be done to know if VO inclusión could affect the taste in comercial size tench.
Lines 332-365: the reduction of EPA-DHA content in absolute values (g x kg-1) showed in table 8 was partially due to the reduction of total lipid content in juveniles fed with high contento f VO. Therefore, EPA-DHA content can increase if total lipid content does.
Lines 332-365: We have try to highlight that, data of lipid content and fatty acid composition of juveniles are only an initial approach on the effects of FO replacement by VO.
Line 368: The fatty acid composition was evaluated on whole-body and not ony in the edible parto of tench because their small size and thus, the number of animals that would have to be slaughtered too high. In other similar studies carried out with small-size fishes is also common perform a whole-body analysis
Conclusión
The last part of discusión was added.
References
References were checked.
Round 2
Reviewer 1 Report
The authors have dealt with my comments well. The manuscript could be accepted for publication in its current form.
Author Response
Thank you for your corrections and suggestions.
Reviewer 2 Report
Thank you for the revisions. The manuscript was improved.
I have minor comment for tables 4-8. Regarding p-values, if it is not significant only two decimals are enough. if it is 0.000 the authors should write it as <0.001. If the p-value is less than 0.01, you can write three decimals, for example 0.003. If the p-value less than 0.05, two decimal places are enough, for example 0.03.
Best wishes,
Reviewer
Author Response
Thank again for your corrections and suggestions. We have changed p-values in table 4-8.
Reviewer 3 Report
Dear Authors,
thank you for considering my previous suggestions on your manuscript, some weaknesses are impossible to fix without repeating these experiments, so I give low value to this manuscript as a general opinion, but it's ready to be published, but needs to be followed by some other studies to better comprehend the data.
Best regards
Author Response
Thanks for your corrections and suggestions. We will try to perform further research on commercial size tench.